# Selective sparsity in Fourier Neural Operator Networks to accelerate Partial Differential Equation solving

## Abstract

Fourier Neural Operators (FNOs) have emerged as a powerful framework for learning solution operators of partial differential equations (PDEs). However, reliance on dense spectral representations leads to high computational cost and limited interpretability. We propose a Spectrally-Sparsified Fourier Neural Operator (SS-FNO) that achieves state-of-the-art accuracy while substantially reducing spectral complexity. Our approach augments each FNO layer with a lightweight sparse selector with a diagonal gating mechanism whose implicit bias under stochastic gradient descent drives many frequency weights toward zero. This induces automatic pruning of uninformative Fourier modes, yielding a compact operator representation that is both efficient and interpretable. We validate SS-FNO on benchmark PDEs, including the Burgers' equation, Darcy flow, and Navier–Stokes equations. Across all cases, SS-FNO matches or exceeds the predictive accuracy of standard FNOs while reducing the number of active frequency modes, reducing the memory footprint and the computation cost. By demonstrating that accurate operator learning does not require dense spectral representations, our work highlights spectral sparsity as a principled path toward scalable and interpretable neural operator models.

## 1 Introduction

A central challenge in computational science is the repeated evaluation of large-scale partial differential equations (PDEs) for varying parameters. PDEs are ubiquitous in physics and engineering, yet applications such as uncertainty quantification, optimal control, or inverse problems in aerodynamics and geophysics may require thousands of PDE solves over high-dimensional parameter spaces. Moreover, these applications often demand fine discretizations, which render traditional solvers computationally expensive and, in many cases, intractable. For example, high-fidelity inverse design of airfoils or seismic subsurface imaging require orders of magnitude more PDE evaluations than what conventional methods can deliver within reasonable time or resource budgets. In such settings, speed is not only desirable but essential: without accelerated PDE solvers, many pressing research questions remain out of reach.

Neural networks have thus become a compelling alternative for data-driven PDE modeling. Physics-informed Neural Networks (PINN) (Raissi et al., 2019) and their variants (Jagtap & Karniadakis, 2020; Zhang et al., 2020), are neural networks trained to model PDEs with a physics-informed loss, can reproduce the results of traditional solvers at a much lower computational cost. One of the disadvantages of using PINNs is that the training dataset needs to be evaluated along fixed grid points, and can therefore be evaluated only at a particular resolution. Changing the resolution requires retraining a new model. This constraint highlights the lack of generalization of PINNs. As an alternative, Lu et al. (2021) have put forward the DeepONet, a neural network architecture capable of learning a neural PDE operator, which can be evaluated on any mesh. Another option is to lift the data to a functional resolution-independent space, as for Graph Neural Operators (Li et al., 2020) and Fourier Neural Operators (Li et al., 2021; Kovachki et al., 2023).

Among these, FNOs have emerged as a landmark method in the field of PDE learning using neural networks, due to their unique architecture that combines the interpretable nature of Fourier coef-

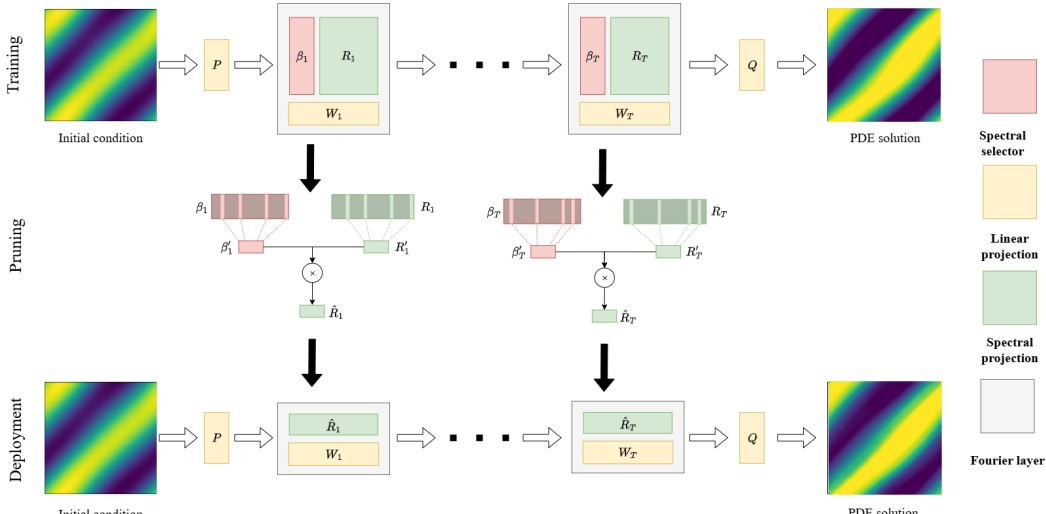

Figure 1: Spectrally-sparse Fourier Neural Operator (SS-FNO) networks: We first train an over-parametrized FNO with a frequency sparsity selector $\beta_t$ for each layer $t$. Once the model is trained, we reduce the frequency selector $\beta_t$ to $\beta'_t$, where we only preserve the most significant values. This allows us to prune the channels of the spectral mixing tensor $R_t$ to produce an equivalent compact $R'_t$. Once compressed, the SS-FNO network can reproduce the forward predictions of the original FNO albeit with a significantly lower number of weights.

ficients with the data-driven approach of neural networks. By operating directly in the Fourier domain, FNOs harness interpretable components such as Fourier coefficients to efficiently capture global dynamics, with the neural network framework providing the flexibility and expressive power needed to model complex, nonlinear behaviors. This combination of interpretability and data-driven adaptability has established FNOs as a foundational approach for data-drive PDE solving.

Despite the fact that many physical phenomena exhibit sparsity in the Fourier domain, this property has not yet been investigated in the context of FNOs. Relevant prior work, includes Sparsified Time-dependent FNO (ST-FNO) Rahman et al. (2024) applied to plasma dynamics, where only the most high-energy modes are retained, with the aim to reduce the computational footprint of the FNO. Exploiting sparsity offers two key advantages: first, it reduces the computational footprint, thus enabling faster PDE inference, which is particularly advantageous in time-critical scenarios such as natural disaster forecasting; second, it reveals which frequency components play a dominant role at each stage of the model. Hence, incorporating sparsity into FNOs can potentially to improve both efficiency and interpretability.

However, up to now, hard regularization constraints have been used to enforce sparsity which essentially modifies the loss function and results in a compromise between accuracy and sparsity. In this work, we propose an alternative: introducing diagonal weight networks as implicit sparse selectors within the Fourier domain. Linear diagonal networks trained with gradient descent have been shown to be implicitly regularized for sparsity (Gunasekar et al., 2018). Pesme et al. (2021) further confirmed this spectral bias with stochastic gradient descent, and has shown that stochasticity in fact increased the sparsity. By using them as Fourier coefficient selectors, we propose to enforce sparse coefficients without any hard regularization. Post-training, the pruned spectral mixing module yields a compact model that preserves accuracy while substantially reducing the number of parameters. Furthermore, this model is also more interpretable since we know the impact of each coefficient on the inference.

In summary, our contributions are: (i) the integration of sparse selectors in the Fourier domain via diagonal weight networks, (ii) a pruning mechanism that yields efficient and interpretable FNOs without accuracy loss, and (iii) validation of our Spectrally-Sparsified FNO (SS-FNO) across canonical PDE benchmarks. Figure 1 provides an overview of the method.

## 2 METHOD

### 2.1 FOURIER NEURAL OPERATOR

Our model builds on the original FNO architecture of Li et al. (2021). In contrast to standard neural networks that learn a learn a mapping (e.g. $f : \mathbb{R}^n \to \mathbb{R}^m$), a neural operator learns a mapping between two functions, such as:

$$\mathcal{G} : \mathcal{A} \to \mathcal{U} \tag{1}$$

where $\mathcal{A}$ is the space of input functions and $\mathcal{U}$ the space of output functions. In the case of PDEs, $\mathcal{A}$ is associated with initial and boundary conditions, whereas $\mathcal{U}$ represents the solutions of the PDE. Given the true operator $\mathcal{G}^\dagger$ and i.i.d. samples $\{a_j, u_j\}_{j=1}^N$ with $u_j = \mathcal{G}^\dagger(a_j)$, the goal is to approximate $\mathcal{G}^\dagger$ by a parametric neural operator $\mathcal{G}_\theta$ with $\theta \in \Theta$:

$$\min_{\theta \in \Theta} \mathbb{E}_a \big[ C(\mathcal{G}_\theta(a), \mathcal{G}^\dagger(a)) \big]. \tag{2}$$

where $C : \mathcal{U} \times \mathcal{U} \to \mathbb{R}$ is a chosen loss function. To compute $C$ over the function spaces, $a_j$ and $u_j$ are discretized with point-wise evaluations. Over the domain $D$, we define the discretizations $D_j = \{x_1, ..., x_n\} \subset D$ such that we have observations $a_j|_{D_j} \in \mathbb{R}^{n \times d_a}$ and $u_j|_{D_j} \in \mathbb{R}^{n \times d_u}$ for each $j$-th pair over $D$.

The neural operator proceeds in three steps:
1. **Lifting**: the input $a$ is mapped into a higher-dimensional space $d_v$ via $v_0(x) = P(a(x))$, where $P$ is a shallow fully-connected network.
2. **Fourier layers**: intermediate features $v_t \to v_{t+1}$ are updated through

$$v_{t+1}(x) = \sigma(W_t v_t(x) + (\mathcal{K}_\phi(a)v_t)(x)), \tag{3}$$

where $W_t$ is a linear projection, $\sigma$ a nonlinear activation, and $\mathcal{K}_\phi$ an integral kernel operator

$$(\mathcal{K}_\phi(a)v_t)(x) = \int_D \kappa_\phi(x, y, a(x), a(y))v_t(y) \, dy, \quad x \in D. \tag{4}$$

Here $\kappa_\phi$ is a neural network with parameters $\phi$.
3. **Projection**: the output is recovered via $u(x) = Q(v_T(x))$ with $Q : \mathbb{R}^{d_v} \to \mathbb{R}^{d_u}$.

The FNO parametrizes $\kappa_\phi$ in the Fourier domain. The Fourier transform $\mathcal{F} : D \to \mathbb{C}^{d_v}$ is defined as:

$$\mathcal{F}f(k) = \int_D f(x)e^{-2i\pi\langle x,k\rangle}\mathrm{d}x, \mathcal{F}^{-1}f(x) = \int_{\mathbb{C}^{d_v}} f(k)e^{2i\pi\langle x,k\rangle}\mathrm{d}k \tag{5}$$

where $k$ is the frequency variable The Fourier integral operator is consequently defined as:

$$(\mathcal{K}_\phi(a)v_t)(x)) = \mathcal{F}^{-1}(R_\phi \cdot (\mathcal{F}v_t))(x) \tag{6}$$

with $R_\phi$ the Fourier transform of a periodic kernel $\kappa : D \to \mathbb{R}^{d_v \times d_v}$. Since $\kappa$ admits a Fourier series, the expansion is truncated at $k_{\max}$, yielding $R_\phi \in \mathbb{R}^{k_{\max} \times d_v \times d_v}$ applied to $\mathcal{F}v_t$.

For further implementation details, we refer the reader to Li et al. (2021).

### 2.2 LINEAR DIAGONAL NETWORKS

Linear Diagonal Networks (LDN) are a simplification of the fully-connected neural networks where the weight matrix is diagonal. More formally, for a LDN of depth $L$ and width $H$, the weights of the network are given as:

$$\mathbf{W} = [\mathbf{w}_1, \mathbf{w}_2, ..., \mathbf{w}_L], \mathbf{w}_l \in \mathbb{R}^H \tag{7}$$

The effective linear predictor $\beta$ of the LDN is hence given as:

$$\beta = P_{\mathrm{diag}(\mathbf{W})} = \mathrm{diag}(\mathbf{w}_1) \odot ... \odot \mathrm{diag}(\mathbf{w}_L) \tag{8}$$

with $\odot$ denoting the Hadamard product.

Soudry et al. (2018) have shown that for separable data, training a neural network with gradient descent implicitly regularizes the weights parameters such that:

$$\mathbf{W}_{\mathrm{GD}}^{\star} = \arg\min ||\mathbf{W}||_2^2 \tag{9}$$

Consequently, Gunasekar et al. (2018) have shown that LDN that when trained using Gradient Descent, LDNs have an implicit penalty on their weights, such that:

$$\min_{\mathbf{W}:P_{\mathrm{diag}}(\mathbf{w})=\beta} ||\mathbf{W}||_2^2 = L||\beta||_{\frac{2}{L}}^{\frac{2}{L}} \tag{10}$$

Thus, the elements of $\beta$ are implicitly regularized by a bridge penalty of $||\beta||_{\frac{2}{L}}$. For $L = 2$, this bridge penalty becomes a Lasso, i.e. a sparse regularizer. Furthermore, Pesme et al. (2021) have shown that using stochastic gradient descent instead of gradient descent enhanced the sparsity penalty.

## 2.3 SPARSE SPECTRUM - FOURIER NEURAL OPERATOR

We introduce our proposed model, the Sparse Spectrum Fourier Neural Operator (SS-FNO), as follows. Building on the original FNO architecture, we introduce in each FNO layer a 2-layer LDN before the channel mixing operation performed by $R_\phi$. Rather than truncating to a small $k_{\mathrm{max}}$, we retain a larger set of Fourier modes, allowing the network to capture higher-frequency content during training. We then use the LDN as a sparse selector, associating each component of $\beta$ to a frequency $k$. By doing so, we impose an implicit sparse regularization on $k$. The architecture is illustrated in Figure 2.

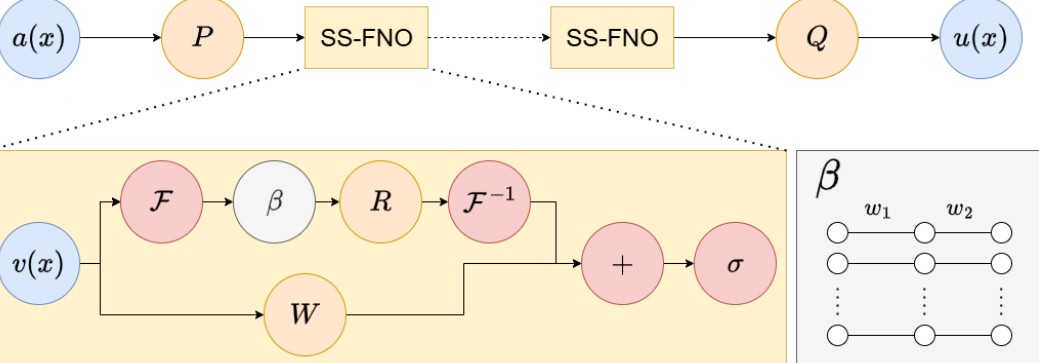

Figure 2: Architecture of the SS-FNO network: the input function $a$ is first lifted into a higher dimension by $P$, after which it is iteratively passed through sparse Fourier layers, and finally is projected back using $Q$. Each of the sparse Fourier layers applies the Fourier transform $\mathcal{F}$, passes it through the sparse selector $\beta$, then a linear transform $R$ before performing the inverse Fourier transform $\mathcal{F}^{-1}$, after which it is combined with a linear projection of the original input with $W$. The sparse selector $\beta$ is implemented as a 2-layer LDN, which is implicitly biased towards sparsity when trained using gradient descent.

Because $k_{\mathrm{max}}$ is larger, training SS-FNO initially incurs higher cost than a standard FNO. After training, however, frequencies with negligible coefficients ($|\beta_k| \ll 1$) are pruned and the remaining values of $\beta$ are folded into $R_\phi$, yielding a compressed model with significantly fewer active modes.

Beyond efficiency, the learned $\beta$ values quantify the contribution of each frequency to the output, providing interpretability into the spectral mechanisms underlying the operator.

## 3 RESULTS

To ensure a fair comparison with the original FNO, we reuse the experimental setup from the original FNO paper from Li et al. (2021). We evaluate FNO and SS-FNO on three benchmark PDEs: the 1-D Burgers' equation, the 2-D Darcy flow problem, and the 3-D Navier-Stokes equations. Each FNO network has four Fourier integral operator layers with ReLU activation and batch normalization. The networks are trained using the Adam optimizer (Kingma & Ba, 2015) with a batch size of 16 over 500 epochs with an initial learning rate of 0.001 that is halved every 100 epochs. The models are compared to each other using the relative error on the test set, in the same manner as the original FNO from Li et al. (2021). The relative error is given as:

$$\text{relative error}(\mathbf{x}, \mathbf{y}) = \frac{||\mathbf{x} - \mathbf{y}||}{||\mathbf{y}||} \tag{11}$$

We also give the computational cost of each model in terms of Mega Floating Point Operations (MFLOPs), where the batch dimension is omitted. The visualizations of $u(x)$ for each experiment is given in the appendix.

### 3.1 1-D: BURGERS' EQUATION

We first consider the 1-D viscous Burgers' equation, a nonlinear PDE modeling wave propagation:

$$\partial_t u(x,t) + \partial_x \frac{u(x,t)^2}{2} = \nu \partial_{xx} u(x,t), \qquad x \in (0,1), t \in (0,1] \tag{12}$$
$$u(x,0) = u_0(x), \qquad x \in (0,1) \tag{13}$$

where $\nu = 0.1$ is the viscosity and $u_0(x)$ is sampled over a Gaussian field with periodic boundary conditions such that $u_0(x) \sim \mathcal{N}(0, 625(-\Delta + 25I)^{-2})$, where $\Delta$ is the Laplacian with periodic boundary conditions and $I$ is the identity operator.

The original FNO is trained with 4 layers, each using 32 modes and $d_v = 64$. For SS-FNO, we begin with 1000 modes per layer and allow the sparse selectors to prune uninformative frequencies. We compare the results at different grid resolutions on Table 1. Despite starting from a significantly larger basis, SS-FNO converges to a compact representation using only about one third as many modes as FNO. Such a reduction leads to more compact spectral weights, reducing the number of parameters by more than half. However, for higher resolutions, the majority of the computation footprint lies in the lifting and residual pass operations. Thus major reductions in computation can only be witnessed at lower resolutions.

We can verify that the LDN truly enforce sparsity by tracking $|\beta|$ during the training process. We plot the distribution of all $|\beta|$ for all layers on Figure 3. Additionally, we can plot the values of $|\beta|$ for each layer for each frequency, as per Figure 4 (in this case SS-FNO resolution 8192). We can thus visualize which frequencies are crucial during the spectral mixing for each layer. For each layer, less frequencies than the default truncation point of the original FNO are necessary to match the relative error. A few examples of the results in the highest resolution can be seen on Figure 5.

### 3.2 2-D: DARCY FLOW PROBLEM

We next evaluate our models on the 2-D Darcy flow equation, an elliptic PDE governing fluid flow through porous media:

$$-\nabla \cdot (a(x)\nabla u(x)) = f(x), \qquad x \in (0,1)^2 \tag{14}$$
$$u(x) = 0 \qquad x \in \partial(0,1)^2 \tag{15}$$

| Model | Resolution | Modes | K Parameters | MFLOPs | Error |
|-------|-----------|-------|--------------|--------|-------|
| FNO | 8192 | 32,32,32,32 | 541 | 143.54 | 0.00223 |
|  | 4096 | 32,32,32,32 | 541 | 74.072 | 0.00229 |
|  | 2048 | 32,32,32,32 | 541 | 39.338 | 0.00220 |
|  | 256 | 32,32,32,32 | 541 | 8.945 | 0.00223 |
| SS-FNO | 8192 | 9,9,9,20 | 209 | 139.431 | 0.00202 |
|  | 4096 | 9,9,9,20 | 209 | 69.963 | 0.00216 |
|  | 2048 | 8,9,10,21 | 213 | 35.241 | 0.00214 |
|  | 256 | 7,9,11,21 | 213 | 4.849 | 0.00230 |

Table 1: Comparison between FNO and SS-FNO for the Burgers' equation. For brevity, Error refers to the relative error, and the number of parameters is given in the thousands. FNO employs a fixed truncation of 32 modes per layer, whereas SS-FNO adaptively selects modes via sparsity. The performances of the models are similar, despite SS-FNO using fewer modes.

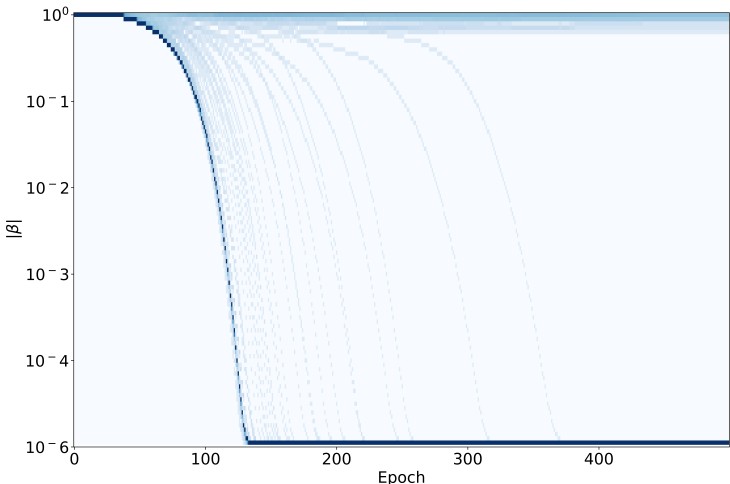

Figure 3: The evolution of the distribution of $|\beta|$ across the training process. The values are rounded up to $10^{-6}$, and the concentration of the color in each bin corresponds to the $\log_{10}$ of the number of samples for better visibility. Most of the values of $|\beta|$ quickly converge to $0$ due the implicit sparsity regularization in LDNs.

where $u(x)$ is the pressure field, $f(x)$ is a fixed forcing term (i.e. $f(x) = 1$), and $a(x)$ is the diffusion coefficient. The coefficient field is generated according to

$$a \sim \mu, \quad \mu = \psi_\# \mathcal{N}\big(0, (-\Delta + 9I)^{-2}\big),$$

with zero Neumann boundary conditions on the Laplacian. Here $\psi : \mathbb{R} \to \mathbb{R}$ is defined point-wise by

$$\psi(z) = \begin{cases} 12, & z \geq 0, \\ 3, & z < 0, \end{cases}$$

so that $a(x)$ takes values 12 or 3 depending on the sign of the Gaussian field.

The baseline FNO is trained with four layers, $d_v = 32$, each using 24 modes; along the first spatial dimension, the modes are doubled to include negative frequencies. SS-FNO is trained in an overparametrized setting with 84 modes per layer before sparsity pruning. We compare the results at different resolutions on Table 2. We observe that SS-FNO achieves comparable accuracy to the standard FNO while retaining only one-third to one-quarter as many modes per layer in both spatial

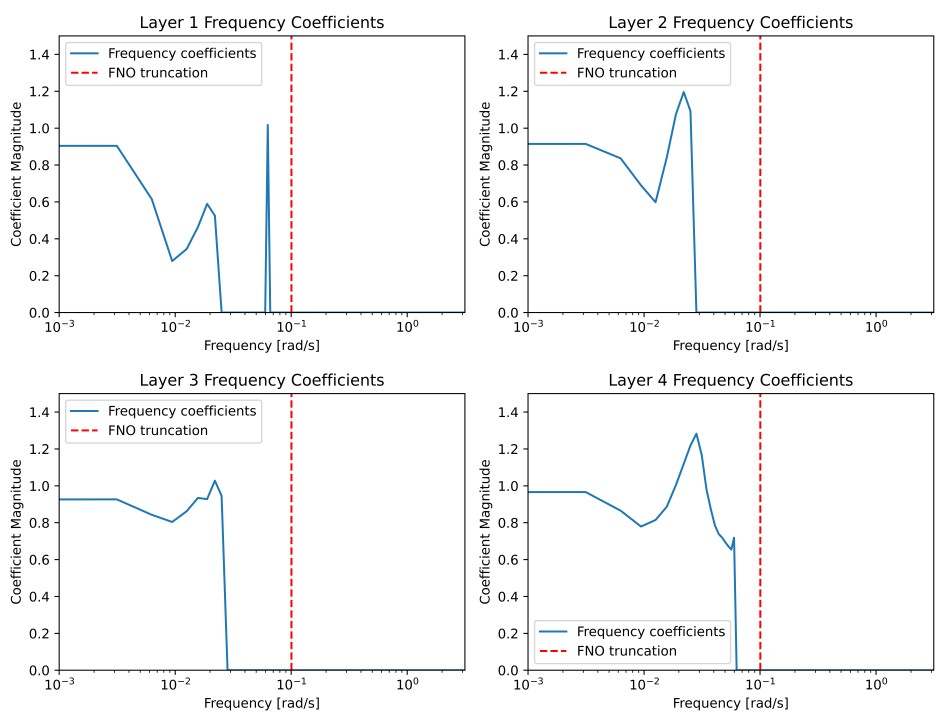

Figure 4: Value of $|\beta|$ for each layer for each frequency (log-scale). For each layer, the number of non-zero modes is less than the truncation threshold from the orignal FNO.

dimensions. Since we reduce the modes along multiple dimensions, the reduction in computation is more notable than for the Burgers' equation case, even for higher resolutions. A few examples of the results in the highest resolution can be seen on Figure 6.

### 3.3 3-D: NAVIER-STOKES EQUATIONS

Our models are further trained on the 3-D (2-D spatial across time) incompressible Navier–Stokes equation over multiple time steps, which are fundamental to the modeling of fluid dynamics. We evaluate both a two-dimensional temporal formulation (FNO-2D) and a full spatio-temporal formulation (FNO-3D). The governing equations in vorticity form are:

$$\partial_t \omega(x,t) + u(x,t) \cdot \nabla \omega(x,t) = \nu \Delta \omega(x,t) + f(x), \qquad x \in (0,1)^2, \ t \in (0,T] \tag{16}$$

$$\nabla \cdot u(x,t) = 0, \qquad x \in (0,1)^2, \ t \in [0,T] \tag{17}$$

$$\omega(x,0) = \omega_0(x), \qquad x \in (0,1)^2 \tag{18}$$

where $\omega(x,t)$ is the vorticity, $u(x,t)$ is the velocity field, and $\nu$ is the viscosity. The initial vorticity is generated according to

$$\omega_0 \sim \mathcal{N}\big(0, \ 7^{3/2}(-\Delta + 49I)^{-2.5}\big),$$

with periodic boundary conditions, where $\Delta$ is the Laplacian and $I$ is the identity operator. The forcing is fixed as

$$f(x) = 0.1\big(\sin(2\pi(x_1 + x_2)) + \cos(2\pi(x_1 + x_2))\big).$$

We implement sparse version of both FNO-2D and FNO-3D. Similarly to the Darcy case, we use 24 modes and $d_v = 32$. The FNO-2D computes the operator step-by-step and can be applied iteratively in time, whereas the FNO-3D computes the entire flow across two spatial and one temporal dimension simultaneously. We train and test our methods on the data made available by Li et al. (2021),

| Model | Resolution | Modes | M Parameters | MFLOPs | Error |
|-------|-----------|-------|--------------|--------|-------|
| FNO | $421 \times 421$ | (48,24),(48,24), (48,24),(48,24) | 4.723 | 850.437 | 0.2441 |
| | $211 \times 211$ | (48,24), (48,24), (48,24), (48,24) | 4.723 | 264.345 | 0.2444 |
| | $141 \times 141$ | (48,24), (48,24), (48,24), (48,24) | 4.723 | 155.535 | 0.2445 |
| | $85 \times 85$ | (48,24), (48,24), (48,24), (48,24) | 4.723 | 99.647 | 0.2445 |
| SS-FNO | $421 \times 421$ | (6, 4),(20, 13), (16, 11), (8, 5) | 1.694 | 787.983 | 0.2470 |
| | $211 \times 211$ | (16, 11), (18, 11), (28, 17), (12, 8) | 3.566 | 206.815 | 0.2472 |
| | $141 \times 141$ | (6, 6), (23, 14), (18, 11), (7, 4) | 1.940 | 94.023 | 0.2597 |
| | $85 \times 85$ | (6, 5), (18, 11), (17, 11), (7, 4) | 1.528 | 36.560 | 0.2480 |

Table 2: Comparison between FNO and SS-FNO for the Darcy flow problem. For brevity, Error refers to the relative error, and the number of parameters is given in the millions. Each tuple element represents the number of modes along each spatial dimension. In the FNO the modes are truncated at a fixed value, while in the SS-FNO each layer has a custom set of modes. The performances of the models are similar, despite SS-FNO using fewer modes.

where $T = 10$ and $\nu = 10^{-3}$. Table 3 reports the comparison. The benefits of our method are most pronounced in the 3-D spatiotemporal case, where the number of parameters is vastly reduced, and the computation footprint is reduced by a third. A few examples of the results can be seen on Figure 7.

| Model | Resolution | Modes | M Parameters | GFLOPs | Error |
|-------|-----------|-------|--------------|--------|-------|
| FNO-2D | $256 \times 256$ | (48, 24), (48, 24), (48, 24), (48, 24) | 4.723 | 3.571 | 0.0223 |
| FNO-3D | $256 \times 256$ | (48, 48, 6), (48, 48, 6), (48, 48, 6), (48, 48, 6) | 94.376 | 4.077 | 0.0041 |
| SS-FNO-2D | $256 \times 256$ | (26, 10), (26, 11), (24, 10), (62, 31) | 2.778 | 3.096 | 0.0199 |
| SS-FNO-3D | $256 \times 256$ | (8, 10, 2), (12, 10, 2), (12, 12, 2), (18, 18, 6) | 2.700 | 2.923 | 0.0045 |

Table 3: Comparison between FNO and SS-FNO for the Navier-Stokes equations. For brevity, Error refers to the relative error, and the number of parameters is given in the millions. Each tuple element represents the number of modes along each dimension. In the FNO the modes are truncated at a fixed value, while in the SS-FNO each layer has a custom set of modes. SS-FNO achieves similar accuracy to FNO while adaptively pruning many modes, with the benefit most pronounced in the 3D spatio-temporal case, where a pruning in all three dimensions leads to a drastic reduction in the number of parameters.

## 4 CONCLUSION

We introduced the Spectrally-Sparsified Fourier Neural Operator (SS-FNO), a variant of FNO that incorporates implicit sparsity through diagonal selectors in the Fourier domain. Across canonical benchmarks, including Burgers' equation, Darcy flow, and Navier-Stokes dynamics, SS-FNO matches or surpasses the accuracy of standard FNOs while requiring substantially fewer active modes. This results in more compact models with faster inference and improved interpretability. While our work leads to an incremental improvement of the FNO, it also introduces the novel concept of relying on implicit regularizations to improve existing architectures. Our findings highlight the practical value of implicit regularization: biases that emerge naturally under gradient-based training can be leveraged to design architectures that are not only theoretically appealing but also scalable and effective in practice. Spectral sparsity therefore provides a principled route toward efficient and interpretable neural operator models for PDEs.

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

## A APPENDIX

### A.1 PARAMETER AND FLOPS

We list the number of parameters for each component of the model. We only include parameters used for evaluation, excluding parameters such as batch normalization weights and $\beta$ which are only used during training.

- Lifting $P$: $d_a \cdot d_v$
- Projection $Q$: $d_v \cdot d_u$
- Skip connection $W$: $d_v \cdot d_v$
- Spectral mixing $R$: $k_{\max} \cdot d_v \cdot d_v$ for 1-D, $k_{\max,x} \cdot k_{\max,y} \cdot d_v \cdot d_v$ for 2-D, $k_{\max,x} \cdot k_{\max,y} \cdot k_{\max,t} \cdot d_v \cdot d_v$ for 3-D

For the FNO, the total number of parameters $\theta_P$ is given as:

$$\theta_{\text{FNO}} = \theta_P + \theta_Q + 4(\theta_W + \theta_R) \tag{19}$$

For the SS-FNO, the values of $k$ vary from layer to layer, yielding $\theta_R$ for different dimensions for each layer. We calculate the number of parameters separately for each layer before summing them together.

Additionally, we also list the FLOPs for each operation (assuming a batch dimension of 1). We also make the assumption that the FFT can be computed up to $k_{\max}$ for any time resolution during deployment.

For the 1-D case:

- Lifting $P$: $N \cdot d_a \cdot d_v$
- Projection $Q$: $N \cdot d_v \cdot d_u$
- Skip connection $W$: $N \cdot d_v \cdot d_v$
- FFT $\mathcal{F}$: $5\, k_{\max} \cdot \log_2(k_{\max}) \cdot d_v$
- iFFT $\mathcal{F}^{-1}$: $5\, k_{\max} \cdot \log_2(k_{\max}) \cdot d_v$
- Spectral mixing $R$: $8\, k_{\max} \cdot d_v \cdot d_v$
- Point-wise operations $\sigma$ and $+$: $N \cdot d_v$

For the 2-D case:

- Lifting $P$: $N_1 \cdot N_2 \cdot d_a \cdot d_v$
- Projection $Q$: $N_1 \cdot N_2 \cdot d_v \cdot d_u$
- Skip connection $W$: $N_1 \cdot N_2 \cdot d_v \cdot d_v$
- FFT $\mathcal{F}$: $10\, k_{\max,x} \cdot k_{\max,y} \cdot \log_2(k_{\max,x} \cdot k_{\max,y}) \cdot d_v$
- iFFT $\mathcal{F}^{-1}$: $10\, k_{\max,x} \cdot k_{\max,y} \cdot \log_2(k_{\max,x} \cdot k_{\max,y}) \cdot d_v$
- Spectral mixing $R$: $8\, k_{\max,x} \cdot k_{\max,y} \cdot d_v \cdot d_v$
- Point-wise operations $\sigma$ and $+$: $N_1 \cdot N_2 \cdot d_v$

For the 3-D case:

- Lifting $P$: $N_1 \cdot N_2 \cdot T \cdot d_a \cdot d_v$
- Projection $Q$: $N_1 \cdot N_2 \cdot T \cdot d_v \cdot d_u$
- Skip connection $W$: $N_1 \cdot N_2 \cdot T \cdot d_v \cdot d_v$
- FFT $\mathcal{F}$: $15\, k_{\max,x} \cdot k_{\max,y} \cdot k_{\max,t} \cdot \log_2(k_{\max,x} \cdot k_{\max,y} \cdot k_{\max,t}) \cdot d_v$
- iFFT $\mathcal{F}^{-1}$: $15\, k_{\max,x} \cdot k_{\max,y} \cdot k_{\max,t} \cdot \log_2(k_{\max,x} \cdot k_{\max,y} \cdot k_{\max,t}) \cdot d_v$
- Spectral mixing $R$: $8\, k_{\max,x} \cdot k_{\max,y} \cdot k_{\max,t} \cdot d_v \cdot d_v$
- Point-wise operations $\sigma$ and $+$: $N_1 \cdot N_2 \cdot T \cdot d_v$

## A.2 EXAMPLE PLOTS

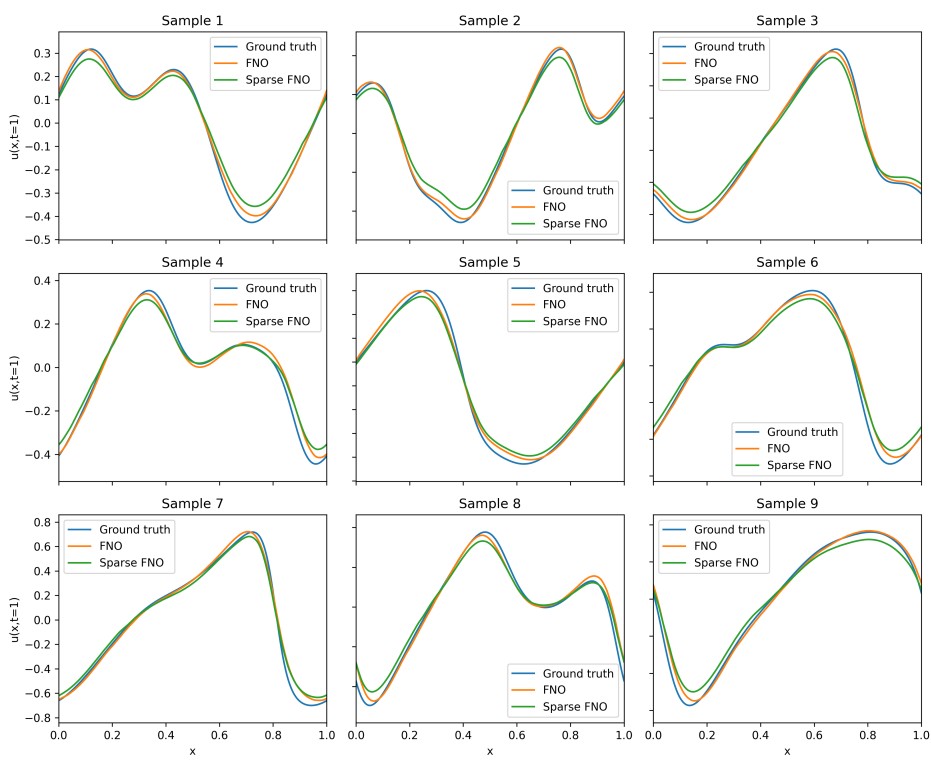

Figure 5: Comparison of the FNO and SS-FNO versus the ground truth for the Burgers' equation.

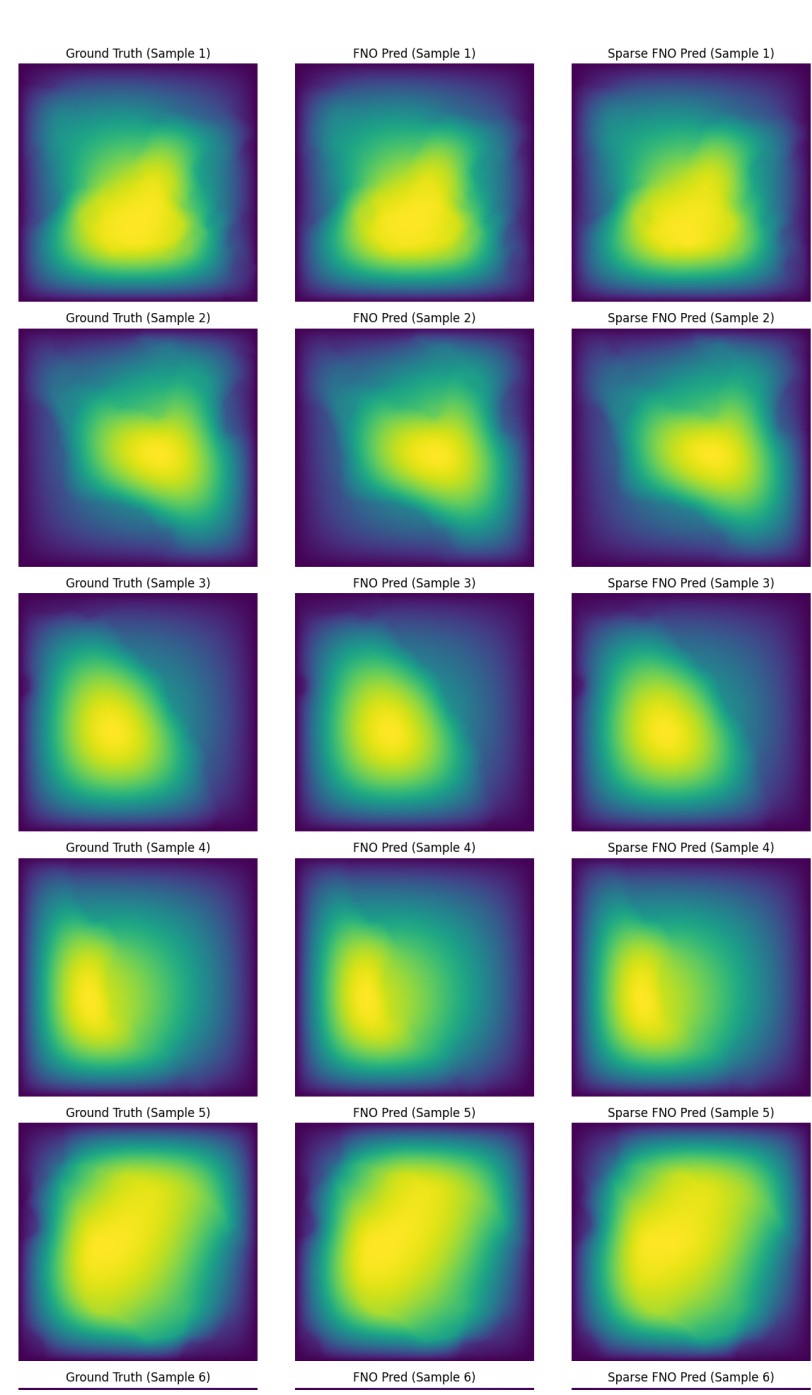

Figure 6: Comparison of the FNO and SS-FNO versus the ground truth for the Darcy flow problem. We display the solutions for different initial conditions.

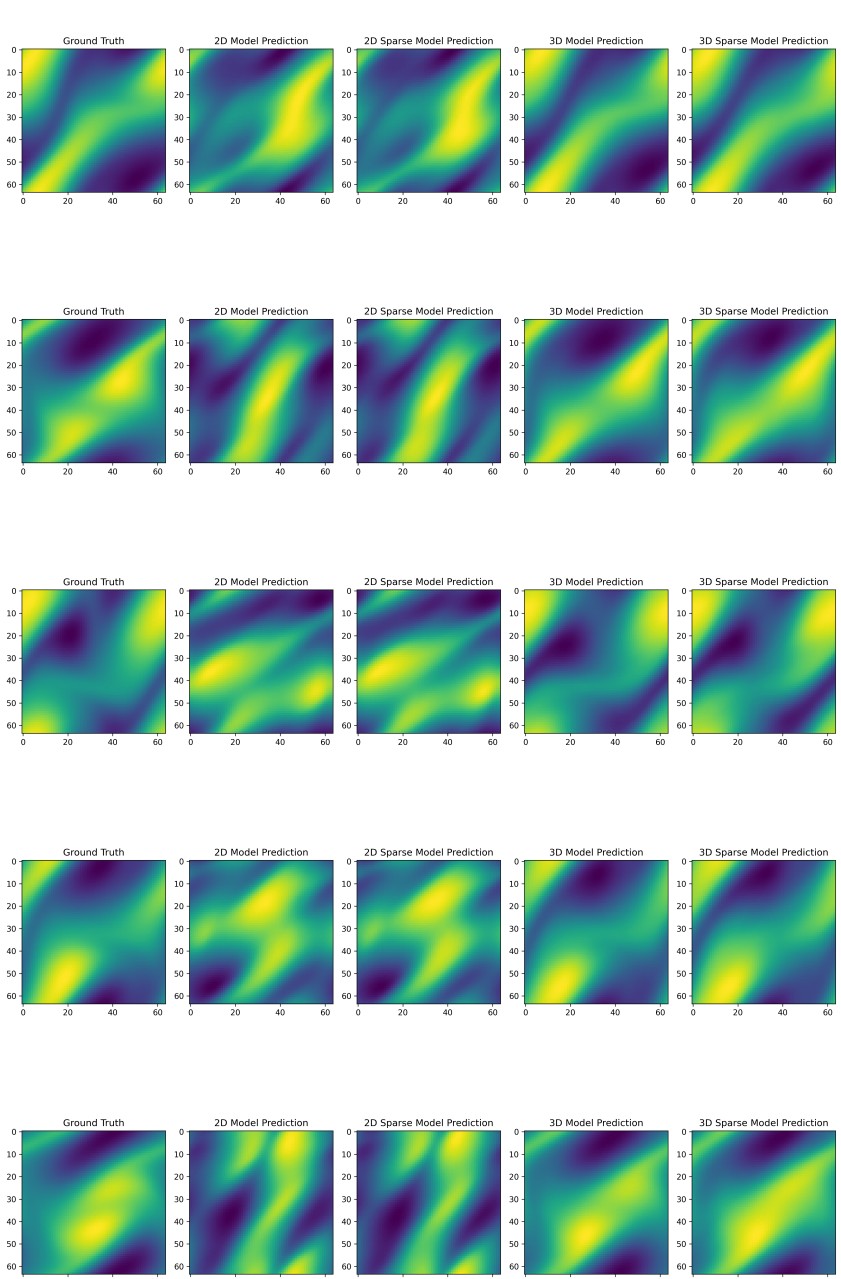

Figure 7: Comparison of the FNO and SS-FNO versus the ground truth for the Navier-Stokes equation. We display the solutions for different initial conditions, and display the 15th step of the forward prediction.

