# OpenReview forum: "Selective sparsity in Fourier Neural Operator Networks to accelerate Partial Differential Equation solving"
_ICLR.cc/2026/Conference — Submitted to ICLR 2026_

### Official Review · Reviewer_zAqe · 2025-10-26

**Soundness:** 2
**Presentation:** 2
**Contribution:** 2
**Rating:** 2
**Confidence:** 5

**Summary:**

This paper introduces the Spectrally-Sparsified Fourier Neural Operator (SS-FNO), which addresses an important limitation of standard FNOs: their reliance on dense spectral representations that leads to high computational costs. The key innovation is the integration of lightweight sparse selectors using 2-layer Linear Diagonal Networks (LDNs) in the Fourier domain. These diagonal networks leverage an implicit bias toward sparsity during stochastic gradient descent training, automatically pruning uninformative frequency modes without requiring explicit regularization penalties. The authors validate SS-FNO on three benchmark PDEs Burgers' equation, Darcy flow, and Navier-Stokes equations demonstrating that it matches or exceeds standard FNO accuracy while substantially reducing the number of active frequency modes, parameters, and computational cost.

**Strengths:**

The core idea here is actually pretty clever using diagonal linear networks to implicitly induce sparsity in the Fourier domain without explicit regularization is neat, and I appreciate that the authors grounded this in existing theory about implicit biases in gradient descent. The experimental setup is solid; they follow the original FNO protocol exactly, which makes the comparisons fair, and they test on three different types of PDEs (Burgers, Darcy, Navier-Stokes) across multiple resolutions. The results do show consistent parameter reduction, especially dramatic in the 3D Navier-Stokes case where they cut parameters by like 70%. I also liked Figure 4 showing which frequencies get selected that's genuinely useful for interpretability and could help us understand what the model is actually learning about these physical systems.

**Weaknesses:**

Honestly, the actual computational gains are pretty underwhelming in several cases. Look at Table 1 for Burgers equation at the highest resolution, they barely save any FLOPs (139.43 vs 143.54, like 3% savings) despite using way fewer modes. The authors hand-wave this away by saying most computation is in the lifting layers, but that's kind of a big deal! If I'm a practitioner trying to speed up my PDE solver, I care about actual wall-clock time, not just the number of parameters. They don't report any training time comparisons either, which seems like a major omission given that they start with way more modes (1000 vs 32). How much longer does training take? How much more memory do I need? These are critical practical questions that aren't answered. Also, the accuracy improvements are marginal at best in many cases SS-FNO actually does slightly worse (look at Table 2, several cases show higher error for SS-FNO). So I'm left wondering: what's the actual benefit here? Slightly fewer parameters but similar or sometimes worse accuracy and unclear computational savings?

The paper also feels incomplete in several ways. The pruning procedure is barely explained; what threshold do they use for "negligible" coefficients? How do they fold β into R? Is there any fine-tuning after pruning? These details matter for reproducibility. There's also basically no analysis of *why* certain frequencies are selected or what this tells us about the PDEs. They claim interpretability as a major benefit, but beyond showing a plot of which frequencies survived, they don't actually interpret anything. Do these patterns make physical sense? Are they consistent across runs? Do they relate to known properties of these equations? Without this analysis, the interpretability claim feels hollow. And where are the comparisons to other sparsification methods? Just saying "we don't use hard regularization" isn't enough they should compare against L1 regularization, magnitude pruning, or other standard techniques to show their implicit approach is actually better. As it stands, I'm not convinced this is a significant enough improvement over standard FNO to warrant publication at a top venue.

**Questions:**

Some questions I had in mind!

1. **Can you provide actual wall-clock training and inference times?** You mention that SS-FNO starts with many more modes (like 1000 vs 32), which must significantly increase training time and memory usage. How long does it actually take to train SS-FNO compared to standard FNO? And more importantly, what are the actual inference speedups in wall-clock time, not just FLOPs? Because looking at Table 1, the FLOP savings seem pretty minimal in some cases, so I'm curious if there's any real-world speed benefit.

2. **How exactly does the pruning work, and how sensitive are results to this choice?** The paper is really vague about the post-training pruning procedure. What specific threshold do you use to decide which coefficients are "negligible"? Is it a fixed value like |β| < 0.01, or a percentile-based cutoff? Do you prune layer-by-layer or globally? And critically, do you do any fine-tuning after pruning? I'm worried that a lot of the performance might depend on these unspecified details.

3. **Why not compare against other sparsification methods?** You only mention ST-FNO as related work, but there are tons of ways to induce sparsityL1 regularization, magnitude pruning, lottery ticket methods, etc. Can you show that your implicit regularization approach actually outperforms these simpler baselines? Without this comparison, it's hard to evaluate whether the added complexity of the diagonal networks is actually necessary.

4. **Can you actually interpret the learned sparsity patterns?** You claim interpretability as a major contribution, but Figure 4 just shows which frequencies survived without any analysis of *why*. Do the selected frequencies correspond to physically meaningful scales in these PDEs? For example, in Navier-Stokes with viscosity 10^-3, do the pruned high frequencies align with scales below the Kolmogorov microscale? Are the patterns consistent across different training runs, or do you get different sparsity structures each time?

5. **Why does SS-FNO sometimes perform worse than standard FNO?** In several experiments (like Darcy flow in Table 2), SS-FNO actually has higher error than FNO. What's going on there? Is this just noise, or is there something about the sparse selection that's hurting performance in certain cases? And given that the accuracy is often similar or worse while training is presumably more expensive, what's the actual value proposition here?

---

> ### Author Response · Authors · 2025-12-02
> **Rebuttal zAqe**
>
> Thank you for reviewing our paper, and for showing interest in our approach. We do agree that the decrease in computation is underwhelming for some of the models put forward.
>
> Regarding the questions:
>
> - 1. We can provide the training times for the training, as per the comments for the other reviewers:
>     - The training time is indeed longer for our method, but is still more or less within the same order of magnitude of the original FNO. We report the following average training epoch on a NVIDIA H100 GPU as follows (FNO/SS-FNO):
>         - Burgers' equation: 0.50s / 0.53s Darcy flow: 7.2s / 12.3s
>         - Navier-Stokes 2D: 3.75s / 6.26s
>         - Navier-Stokes 3D: 2.67 vs. 9.75s
>     The times for the inference times cannot be provided, due to the lack of a specialized sparse FFT method for GPUs.
> - 2. We do agree that our description of the pruning is vague. As the reviewer has correctly guess, the channels are pruned according to a threshold value of the norm of the value of the estimator, typically $|\beta|<0.001$ for the models in our work. The pruning is done globally, and there is no finetuning performed after the pruning.
> - 3. As per another reviewers questions: "The implementation of implicit L1 regularization would indeed have provided an interesting comparison for our method. We do however note that this comparison would not be so straightforward to implement. In addition to the introduction of additional hyperparameters to account for (requiring additional ablation studies), the minimization of L1 losses typically requires specialized variants of gradient descent (ISTA, FISTA, etc...) or proxy losses (e.g. SmoothL1 loss). These significant changes to the training process make it more difficult to truly compare both methods." Furthermore, the current reviewer also brings up more naive implementations such as magnitude pruning. Comparisons to such methods would have probably made our model more compelling.
> - 4. We did not verify whether the pruned high frequencies align with scales below the Kolmogorov microscale. We agree that a more thorough study of the interpretability of our model would have been necessary.
> - 5. We suspect that this is due to noise in the training procedure. Repeating the training over multiple seeds would have allowed us to explain these discrepancies. We apologize for the confusion.

---

### Official Review · Reviewer_7tY6 · 2025-10-27

**Soundness:** 1
**Presentation:** 2
**Contribution:** 1
**Rating:** 2
**Confidence:** 4

**Summary:**

The paper introduces SS-FNO (Spectrally Sparsified Fourier Neural Operator), a novel architecture for efficiently solving PDEs. SS-FNO employs a “sparse selector” mechanism that automatically prunes less important frequency components during training, induced by stochastic gradient descent. As a result, SS-FNO produces a more efficient & effective solution than FNO.

**Strengths:**

1. SS-FNO requires fewer active frequency modes, reducing both memory footprint and computational cost.
2. The sparse selector quantifies the importance of each frequency, allowing insight into the model’s mechanisms.

**Weaknesses:**

1. The improvements are incremental, as they build on an established method (FNO) rather than a completely new paradigm. Pruning unimportant channels and frequencies is nothing but a mature technique in computer vision.
2. The SS-FNO demonstrates limited improvement in both efficiency and effectiveness against FNO.
3. Also, only FNO is selected as the baseline, making it hard to evaluate the proposed SS-FNO in the bigger picture.

**Questions:**

See cons

---

> ### Author Response · Authors · 2025-12-01
> **Rebuttal 7tY6**
>
> Thank you for reviewing our work. Regarding the weaknesses cited in the review:
>
> - 1. Our work is incremental with regards to the improvement of the accuracy/efficiency of the original FNO.
> - 2. The effectiveness may be indeed hard to measure, especially considering the lack of specialized GPU functions to put our improvements into practice.
> - 3. The comparison with other methods would indeed strengthen the claims in our paper.

---

### Official Review · Reviewer_qC5z · 2025-11-01

**Soundness:** 3
**Presentation:** 3
**Contribution:** 3
**Rating:** 4
**Confidence:** 5

**Summary:**

This paper introduces Spectrally-Sparsified Fourier Neural Operators (SS-FNOs), which learn to selectively maintain the most relevant Fourier modes during training through sparse selectors. The method achieves comparable accuracy to standard FNOs while significantly reducing the spectral complexity, leading to more compact and interpretable models. Benchmarks on several PDEs demonstrate that SS-FNOs maintain predictive performance with faster inference and improved interpretability.

**Strengths:**

1. The authors evaluate SS-FNO across multiple PDE benchmarks (Burgers’, Darcy flow, and Navier–Stokes), demonstrating comparable accuracy under substantial parameter reduction.
2. The learned spectral weights offer interpretable insights into which frequency components are essential for each task.
3. SS-FNO reduces parameters significantly, making FNOs more compact and potentially deployable for large-scale PDE problems.

**Weaknesses:**

The paper lacks ablation studies that verify whether the learned spectral sparsity truly contributes to the model’s performance. It can compare with Low-frequency and random pruning baseline. Additionally, it would be insightful to compare the learned spectral weights with the true dominant modes from the underlying physics.

**Questions:**

1. The paper reports FLOPs and parameter reductions but does not provide wall-clock inference time. It would strengthen the claim of “faster inference” to include empirical timing results.
2. The proposed method maintains a larger modes during training. Does this lead to higher computational cost or memory usage compared to FNOs? Reporting training time (per epoch) would be informative.

---

> ### Author Response · Authors · 2025-12-01
> **Reviewer qC5z**
>
> Thank you for reviewing our work. We agree that additional ablation studies and comparison with explicit sparsification methods would have made our argument more convincing. Regarding the questions:
>
> - 1. Unfortunately, the decrease in inference cannot be implemented without custom GPU sparse FFT functions, which we have not implemented.
> - 2. The training time is indeed longer for our method, but is still more or less within the same order of magnitude of the original FNO. We report the following average training epoch on a NVIDIA H100 GPU as follows (FNO/SS-FNO):
>     - Burgers' equation: 0.50s / 0.53s
>     - Darcy flow: 7.2s / 12.3s
>     - Navier-Stokes 2D: 3.75s / 6.26s
>     - Navier-Stokes 3D: 2.67 vs. 9.75s

---

### Official Review · Reviewer_HhnP · 2025-11-01

**Soundness:** 2
**Presentation:** 2
**Contribution:** 2
**Rating:** 2
**Confidence:** 5

**Summary:**

This paper introduces the Spectrally-Sparsified Fourier Neural Operator (SS-FNO), a modification of the FNO designed to induce sparsity in the spectral domain. The core of the method is the addition of a lightweight, 2-layer Linear Diagonal Network (LDN) to each Fourier layer, which acts as a "sparse selector" for the frequency modes. The authors leverage the known implicit bias of LDNs trained with stochastic gradient descent, which naturally encourages an L1-like sparsity penalty without requiring an explicit regularization term in the loss function. The workflow involves training an over-parameterized SS-FNO with a large number of modes, allowing the LDN to automatically drive uninformative mode coefficients to zero. This is followed by a post-training pruning step to remove these negligible modes, resulting in a compressed, efficient model for deployment.

**Strengths:**

The primary strength of the paper is its demonstrated ability to significantly compress the FNO architecture. The method achieves a substantial reduction in the number of active Fourier modes across all benchmarks (e.g., from 32 to as few as 9 in the 1D Burgers' case). This compression is achieved with a negligible impact on predictive accuracy. The experimental results show that the SS-FNO consistently "matches or exceeds" the accuracy of the standard FNO. The method for inducing sparsity is theoretically grounded. Instead of relying on a hand-tuned L1 penalty in the loss function, the authors cleverly leverage the implicit bias of gradient descent on Linear Diagonal Networks (LDNs).

**Weaknesses:**

1. The paper's contribution, while effective, feels incremental. The authors themselves state it is an "incremental improvement of the FNO". The core FNO architecture is unchanged, and the paper essentially presents a structured, implicit pruning technique.

2. The experimental validation is weak. The only baseline used is the original FNO from 2021. This is insufficient. As a paper focused on FNO compression, it fails to compare against any other FNO compression or sparsity-inducing techniques. Prominent and highly relevant baselines are missing, such as tensor-factorized FNOs (e.g., TFNO, MG-TFNO), which are known to achieve massive compression and performance gains. Without this comparison, it is impossible to assess if this complex, implicit bias-based method is superior to more direct approaches like tensor decomposition or other pruning strategies.

3. The paper's claims of computational reduction are not fully convincing.
- Training Cost: The SS-FNO method is more expensive to train than the baseline, as it requires starting with an "overparametrized setting" (e.g., 1000 modes for Burgers' vs. 32 for FNO).
- Inference Cost: The reported MFLOPs/GFLOPs reductions are modest in some high-resolution cases (e.g., 1D Burgers and 2D Darcy) and the paper admits this. More critically, the resulting sparsity patterns are highly irregular (e.g., mode counts of (26, 10), (26, 11), etc.), which are known to map very inefficiently to modern GPU hardware (e.g., tensor cores) and can introduce significant overhead from irregular memory access. The paper provides no wall-clock runtime comparison to verify if the theoretical FLOPs reduction translates to a real-world speedup.

**Questions:**

1. Figure 4 shows that for the Burgers' equation, the SS-FNO automatically learns to keep far fewer modes (e.g., ~10-20) than the baseline FNO's fixed truncation of 32. This suggests the baseline FNO may be poorly tuned. What is the performance of the standard FNO if its truncation hyperparameter kmax is simply set to 20, or the other sparse counts discovered by SS-FNO?

2. The learned sparse mode counts are highly irregular. FNO implementations typically use mode counts that are powers of 2 (or other hardware-friendly numbers) to maximize computational efficiency on GPUs. Do these irregular, sparse-mode tensors actually lead to a practical, wall-clock speedup during inference, or does the overhead from gather/scatter operations and inefficient tensor core usage negate the theoretical FLOPs reduction?

3. The paper's core premise is the benefit of implicit regularization from LDNs over "hard regularization constraints". However, this claim is never substantiated. A critical missing ablation study would compare SS-FNO against a standard FNO trained with a simple, explicit L1 penalty on the mode weights. This would clarify if the complexity of the LDN selector is truly necessary.

---

> ### Author Response · Authors · 2025-12-01
> **Rebuttal HhnP**
>
> Thank you very much for taking the time to thoroughly review our paper. We agree with the assertions that:
> - our work is incremental with regards to the improvement of the accuracy/efficiency of the original FNO.
> - the original FNO actually implements architectures with both 16 and 32 modes. Due to the concentration of the dynamics of Burgers' equation being in the low frequencies (c.f. original paper), the 16-mode model also converges to a similar level of accuracy. Our work mainly shows that this manual tuning can be replaced with implicit regularization.
> - our work only accelerates the FNO from a theoretical perspective. For these gains to translate to an actual acceleration of the computation, we would need to implement specialized FFT algorithm that can scatter/gather sparse modes on the GPU, which we have not implemented.
> - the training cost is obviously much higher for our model than for the original FNO.
>
> Regarding the questions:
> - 1. The original FNO actually implements architectures with both 16 and 32 modes. Due to the concentration of the dynamics of Burgers' equation being in the low frequencies (c.f. original paper), the 16-mode model also converges to a similar level of accuracy. Our work mainly shows that this manual tuning can be replaced with implicit regularization.
> - 2. As answered previously, our work only results in a theoretical acceleration of the original FNO. Specialized GPU functions are necessary for these gains to materialize.
> - 3. The implementation of implicit L1 regularization would indeed have provided an interesting comparison for our method. We do however note that this comparison would not be so straightforward to implement. In addition to the introduction of additional hyperparameters to account for (requiring additional ablation studies), the minimization of L1 losses typically requires specialized variants of gradient descent (ISTA, FISTA, etc...) or proxy losses (e.g. SmoothL1 loss). These significant changes to the training process make it more difficult to truly compare both methods.

---

### Meta-Review · Area_Chair_ZeNR · 2026-01-06

**Summary:**

The reviewers generally indicate that:
1. The contribution is incremental, introducing a small change to the weight parameterization of the FNO.
2. The evaluation datasets are too few and too simple.
3. The evaluation baselines are outdated missing similar, parameter reducing techniques such as TFNO and F-FNO.
4. The computational advantage during inference time of the model is very marginal compared to a standard FNO.
5. Training times are longer for the proposed method compared to a standard FNO.

**Reviewer Concerns:**

The authors were unable to sufficiently address the majority of reviewer concerns. They provided per epoch training times for their method compared to a standard FNO, showing it to not be significantly worse. All other concerns remain.

**Reviewer Scores:**

All reviewer scores would remain the same.

---

### Decision · Program_Chairs · 2026-01-26

Reject